# Seasonal variations in haematological and biochemical parameters of healthy Gambian adults: Retrospective study 2018–2022

Mustapha Dibbasey[1,2]*, Mamudou Dahaba[1], Francess Sarfo[1], Rosyna Begum[1], Mustapha Kanteh[1], Nyima Sumareh[1], Mustapha Bakare[1], Solomon Umukoro[1], Alfred Amambua-Ngwa[1]*

**1** Medical Research Council Unit The Gambia at the London School of Hygiene and Tropical Medicine, Banjul, The Gambia, **2** Department of Biochemistry, Cell and Molecular Biology, School of Biological Science, University of Ghana, Accra, Legon, Ghana

* lshmd4@lshtm.ac.uk, mdibbasey@mrc.gm (MD); Alfred.Ngwa@lshtm.ac.uk (AA-N)

**Data Availability Statement:** The data for the study is fully available and submitted with manuscript.

## Abstract

The objectives of this study were to determine the presence and effect of seasonal variations and provide insights into trend from 2018 to 2022 in a comprehensive set of routine haematological indices and biochemical measurements in Gambian adults with no known underlying health condition. We retrieved five years of data from an electronic database and analysed 493 full blood counts and 643 biochemical data from different individuals. In this study, we focused on data from individuals with no known underlying health condition who visited the clinical diagnostic laboratory for routine medical examinations or assessments. Our study found a positive association between seasonality (wet season as the reference) and Hb (HB: 0.014(0.015), P<0.05), White blood cells (WBC) (WBC: 0.243(0.163), p = 0.0014), and neutrophils (neutrophils: 0.271(0.131), P<0.05) with exception to red blood cells (RBC) (RBC: - 0.184(0.061), P< 0.003) that showed negative association. Despite the association, the seasonal effects on our derived reference intervals for haematological indices and biochemical measurements from wet season to dry season were not statistically significant (P>0.05). In addition, we observed in our heatmap result that some laboratory parameters, including HB, RBC, haematocrit (HCT), urea, liver enzymes, and potassium, showed seasonal variation patterns throughout the year, with median levels being normal to slightly low during the dry season and normal to high during the wet season. We also found no significant difference (P>0.05) among the median values for all parameters from 2018 to 2022. Additionally, aspartate aminotransferase (AST), and alanine aminotransferase (ALT) parameters showed a consistent declining trend from 2018 to 2022. Our study found no seasonal effects on the derived reference intervals of haematological indices and biochemical measurements. However, we observed changes in patterns for certain parameters particularly HB, RBC, liver enzymes, and potassium based on seasonality.

**Funding:** This project received no specific funding for this work.

**Competing interests:** The authors have declared that no competing interests exist.

## Introduction

Routine haematological and biochemical parameters are essential for assessing an individual's health and diagnosing various medical conditions [1–4]. Seasonal and yearly variations in these parameters can be influenced by environmental factors such as temperature, humidity, and rainfall. Changes in environmental conditions can impact an individual's dietary preferences and physical activity levels [5–8], which in turn affect their metabolic processes and overall health. Additionally, certain diseases, including respiratory infections, malaria, and parasitic infestations, often exhibit seasonal patterns that can impact haematological and biochemical parameters, particularly total white cell count and differentials. Moreover, the COVID-19 pandemic has caused significant lifestyle changes, such as physical inactivity and limited access to a healthy diet, which can also affect the general health of populations or countries [9–11].

Investigating seasonal variations and yearly trends in routine parameters over the past five years is of clinical and epidemiological significance. Few studies conducted in sub-Saharan Africa have considered the seasonal effects on haematological and biochemical parameters reference intervals and have shown to affect the reference intervals [1–3,12]. However, an earlier study conducted in Rwanda, Uganda, and Zambia reported minimal impact of seasonality on adult clinical laboratory parameter values in these populations, suggesting that seasonal variation may not be important in evaluating adult clinical laboratory parameters [13]. Despite these contradictory findings, seasonal influences on haematological and biochemical parameters can potentially impact the interpretation of laboratory results, disease diagnosis, and patient care management. In the ongoing country-tailored reference interval study project, this retrospective study has the potential to contribute to refining geographically and genetically tailored reference ranges for routine parameters in the context of seasonality, ensuring more accurate and context-specific interpretation of laboratory results.

This study aimed to evaluate the presence and extent of seasonal variations in a comprehensive set of haematological and biochemical routine parameters in Gambian adults with no known underlying health condition and their effects on our defined reference intervals. We also examined the trending pattern of the main haematological and biochemical parameters over the past five years. By utilizing the retrospective data captured in the electronic medical record system of the Medical Research Council Unit The Gambia at LSHTM (MRCG@LSHTM) clinical laboratories, we provided a description of these variations and their potential clinical implications.

## Method

### Study setting

The study was conducted in the ISO15189:2012 and good clinical laboratory practice accredited clinical diagnostic laboratory hosted at the Kuyateh Building at MRCG@LSHTM, which is located in the Kanifing municipality in the greater Banjul area of The Gambia [14–16]. The diagnostic laboratory is responsible for providing standard and quality-oriented diagnostic services to MRCG@LSHTM staff, clinical trial projects, as well as government and private hospitals. Staff employed by MRCG@LSHTM underwent routine medical examinations at the clinical diagnostic laboratory, and their haematological and biochemical results data were captured in an electronic medical record system database.

### Study design and data extraction

This retrospective study retrieved haematological full blood count results data and biochemical results data from 1st January 2018 to 31st December 2022 from the MRCG@LSHTM electronic

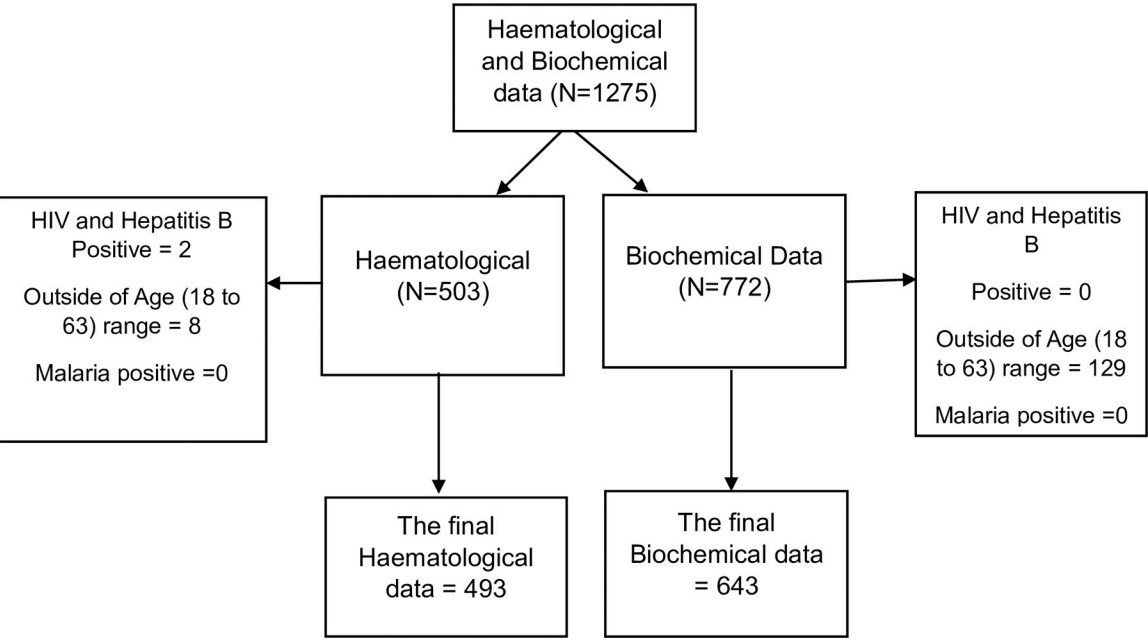

**Fig 1. Flow chart demonstrating the data retrieval process and excluded records.**

medical record system database. In this study, we focused on data from individuals with no known underlying health condition who visited the clinical laboratory department for routine medical examinations and have their haematological and biochemical tests performed. After completing the retrieval process and conducting data cleaning, a total of 493 haematological full blood count results and 643 biochemical results were retrieved from the electronic database (Fig 1 and S1 Data). Out of the 643 biochemical results, 150 were identified as lacking corresponding haematological full blood count results in the electronic database. However, these 150 results were still included in the final analyses to increase the sample size. The extracted haematological full blood count dataset included Hb, RBC, platelets, and WBC and differentials results. The extracted biochemical dataset comprised serum Gamma-glutamyl transpeptidase (GGT), Sodium, Potassium, Urea, Creatinine, AST, ALT, Total Protein (TP), Albumin (ALB), and Globulin results.

During the data extraction, we considered the age range of 18–63 years, in accordance with the employment age range outlined in MRCG@LSHTM policies. Based on clinical information available in the electronic system, we excluded data from the study if the electronic medical records showed positive serological results for human immunodeficiency virus and hepatitis B virus as well as positive malaria and pregnancy test (Fig 1).

## Ethics approval and consent to participate

This study was approved by the Gambia Government/MRCG Joint Ethics Committee (S1 Text). Since all data retrieved from the database were anonymized and coded, informed consent was waived by the Ethics Committee. All methods were performed in accordance with the Declaration of Helsinki.

## Statistics

The data were retrieved in Microsoft Excel and analysed using the R package to obtain summary statistics, including the mean, median, and plots. The Shapiro test was validated by Q-Q

plot, and a normal distribution bell-shaped curve was used to study the distribution pattern of our data across the parameters. A box plot was used to assess the presence of outliers in datasets. Reference intervals were determined based on the 2.5th and 97.5th percentiles after eliminating outliers in our data using interquartile range calculations (25th—1.5 * IQR; 75th—1.5 * IQR). The comparison between the dry and wet season medians was performed by nonparametric Wilcox Rank test in R. Multivariate analysis using a generalized linear modelling approach in R was used to model the association between explanatory variables (season, age, gender and year) and haematological and biochemical parameters. We further compared the medians of 2018 to 2021 using the Dunn test in R to generate Kruskal-Wallis chi-square and p-value results. P value (P) of less than 0.05 was used to determine statistical significance.

## Results

From this study, 493 haematological full blood count data (N = 493) from haematology laboratory department that comprises haematological indices such as RBC, HB, PLT, and WBC and differentials were analysed (Table 1). Additionally, 643 biochemistry laboratory data retrieved from clinical biochemistry laboratory department that comprises biochemical parameters such as liver enzymes, sodium, potassium, urea, and creatinine were analysed (Table 1).

Over the five-year period from 2018 to 2022, the seasonality (dry and wet seasons) was defined based on the visit date for sample collection at the clinical laboratory department. The wet season was represented by visits from June to October, while the dry season was represented by visits from November to May. Based on the visit data, 55% of the full blood count data and 58% of the biochemistry laboratory data were generated in the dry season (Table 1).

**Table 1. Summary table of characteristics of sample population.**

| Explanatory variables | Characteristics | Summary N(%) |
|---|---|---|
| **Haematology laboratory** | | |
| Season (N = 493) | Dry (n) | 271 (55.0) |
| | Wet (n) | 222 (45.0) |
| Gender (N = 493) | Male (n) | 300 (60.85) |
| | Female (n) | 193 (39.15) |
| Dry Season (N = 271) | Male (n) | 175 (64.58) |
| | Female (n) | 96 (35.42) |
| Wet Season (N = 222) | Male (n) | 125 (56.31) |
| | Female (n) | 97 (43.69) |
| **Biochemistry Laboratory** | | |
| Season (N = 643) | Dry (n) | 373(58.01) |
| | Wet(n) | 270(41.99) |
| Dry Season (N = 373) | Male(n) | 183(49.06) |
| | Female (n) | 190(50.94) |
| Wet Season (N = 270) | Male(n) | 148(54.81) |
| | Female(n) | 122(45.19) |
| Gender (N = 643) | Male (n) | 339 (52.72) |
| | Female(n) | 304(47.28) |
| **Age profile of the two datasets retrieved** | | |

| Datasets | Age Range | Median | Mean |
|---|---|---|---|
| Haematological Dataset | 18–63 | 28.0 | 30.5 |
| Biochemical Dataset | 18–63 | 33.0 | 36.2 |

Seasonal and yearly variations in the routine haematological and biochemical parameters

Unlike the biochemistry laboratory data, which showed almost equal gender proportions, there was a gender disparity in the haematological laboratory data, with 60% of the routine medical examination full blood count data came from the male population (Table 1).

The age range for both haematological and biochemistry datasets was 18 to 63 years. The mean age of the haematological dataset was 31 years, and the median age was 28 years (Table 1). For biochemistry data, the mean age was 36 years, and the median age is 33 years. However, there was no significant difference between the mean and median ages of the two datasets (Table 1).

### Seasonal and yearly effects and reference intervals

To assess the seasonal effects on the haematological indices and biochemical parameters, we analysed association between seasonality (i.e. whether the sample was collected in wet or dry season), and the haematological indices and biochemical parameters (Table 3). Using generalised linear modelling (GLM), we found that seasonality was associated with various haematological indices (Hb: 0.014(0.015), P<0.05; RBC: -0.184(0.061), P<0.003; PLTs: 17.47 (6.86), p<0.01; WBC: 0.243(0.163), p = 0.0014; neutrophils: 0.271(0.131), P<0.05; and lymphocytes: -0.003(0.066), P = 0.01). However in the biochemical dataset, the biochemical measurements were not affected by the explanatory variables of dry and wet seasons.

In the haematological dataset, age was found to have an association with certain parameters, including haemoglobin (0.028(0.011); P = 0.011), platelets (-0.918(0.44): P = 0.039). In the biochemical dataset, only creatinine (1.202(0.472); P<0.01), and total protein (-0.117(0.048); P<0.05) were found to have an association with age (Table 2). Furthermore, several haematological indices, except for monocytes and eosinophils, were associated with gender. In the biochemical dataset, only sodium (0.680(0.323); P = 0.04) was found to have significant association with gender.

Next, we assessed the association of the year of visit on various parameters using the GLM approach (Table 3). We found significant associations between the year of visit and haematological parameters such as haemoglobin (0.156(0.079): P<0.05), RBC (0.080(0.034): P<0.05), and platelets (7.861(3.16); P<0.05). In the biochemical dataset, biochemical measurements such as sodium (-0.93(0.31); P<0.05), potassium (-0.107(0.041); P = 0.013), albumin (-0.89 (0.43); P = 0.04), and globulin (1.33(0.50); P = 0.01) were found to be associated with the year explanatory variable.

We determined reference intervals nonparametrically based on the 2.5th and 97.5th percentiles for both haematological indices and biochemical measurements (Table 3). To assess the effects of seasonality, we compared the median values between the dry season and the wet season for each year from 2018 to 2022. Although the differences were not statistically significant across all haematological indices (P>0.05), the median values for platelets, as well as WBC and differentials, were slightly higher in the wet season compared to the dry season. Conversely, the median values for Hb and RBC were lower in the wet season. Similarly, the median values for all the biochemical measurements were not statistically significant (P>0.05). However, potassium had a slightly higher median in the dry season (median = 4.20) compared to the wet season (median = 4.10).

We also evaluated the effects of seasonality on the reference intervals derived from our haematological and biochemical study datasets. In haematological data, we observed changes in reference intervals from the dry season to the wet season were not significant (P>0.05). However, we observed a slight reduction in the upper limit of reference intervals in the wet season for HB and RBC haematological indices, and a slight increase in the upper limit of WBC and

**Table 2. Association study of explanatory variables season, age and gender, and all the haematological and biochemical parameters.** The association was determined by estimated effects on the median (SE) value of the parameters. P value <0.05; Standard Error of the estimate intercept (SE).

| Laboratory parameters | Season[a]<br>Estimate (Std. Error);<br>P value <0.05 | Age<br>Estimate (Std. Error);<br>P value <0.05 | Gender[b]<br>Estimate (Std. Error);<br>P value <0.05 | Year<br>Estimate (Std. Error);<br>P value <0.05 |
|---|---|---|---|---|
| HB (g/dL) | **0.014(0.015);<br>P<0.05** | **0.028(0.011); P = 0.011** | **2.355(0.126); P<0.0001** | **0.156(0.079): P<0.05** |
| RBCa(106/uL) | **-0.184(0.061);<br>P<0.05** | 0.004(0.003); P = 0.287 | **0.750(0.049); P<0.0001** | **0.080(0.034): P<0.05** |
| PLT (103/uL) | **17.47 (6.86);<br>P<0.01** | **-0.918(0.44): P = 0.039** | **-60.19(6.276); P<0.0001** | **7.861(3.16); P<0.05** |
| WBC (103/uL) | **0.243(0.163);<br>P<0.01** | -0.006(0.011); P = 0.566 | **-0.929(0.158); P<0.0001** | 0.030(0.071) P>0.67 |
| Neutrophils (103/uL) | **0.271(0.131);<br>P<0.05** | -0.009(0.008); P = 0.295 | **-0.624(0.129); P<0.0001** | -0.027(0.05); P = 0.61 |
| Lymphocytes (103/uL) | **-0.003(0.066);<br>P< 0.01** | 0.006(0.004); P = 0.169 | **-0.288(0.0654); P<0.0001** | 0.038(0.03); P = 0.22 |
| Monocytes (103/uL) | -0.004(0.165); P = 0.798 | -0.002(0.001); P = 0.092 | -0.027(0.017); P = 0.11 | **0.037(0.007); P<0.05** |
| Eosinophils (103/uL) | -0.021(0.023); P = 0.359 | -0.001(0.001); P = 0.380) | 0.011(0.023); P = 0.623 | -0.01(0.01); P = 0.139 |
| Sodium<br>(mmol/L) | -0.238(0.330);P = 0.471 | 0.0188(0.013); P = 0.156 | **0.680(0.323); P = 0.04** | **-0.93(0.31); P<0.05** |
| GGT (U/L) | 0.213(7.31);<br>P = 0.977 | 0.227(0.241); P = 0.347 | 10.696(7.27); P = 0.142 | -0.43(1.39); P = 0.76 |
| Potassium<br>(mmol/L) | 0.020(0.039);<br>P = 0.6143 | 0.001(0.001): P = 0.424 | 0.065(0.039); P = 0.098 | **-0.11(0.04); P = 0.013** |
| Urea (mmol/L) | -4.128(0.259);<br>P = 0.112 | 0.064(0.040); P = 0.113 | 1.225(1.301); P = 0.347 | -0.11(0.136); P = 0.42 |
| Creatinine<br>(umol/L) | -18.13(14.88);<br>P = 0.224 | **1.202(0.472); P<0.01** | 28.15(15.23); P = 0.066 | 0.62(2.00); P = 0.76 |
| AST (U/L) | -2.045(2.256);<br>P<0.344 | -0.066(0.069); P = 0.597 | 0.566(0.256); P = 0.798 | -0.56(0.65); P = 0.40 |
| ALT (U/L) | -1.619(2.60);<br>P = 0.534 | 0.031(0.089); P = 0.729 | 1.244(2.846); P = 0.662 | -1.69(1.12); P = 0.14 |
| Albumin (g/dL) | -1.663(0.863);<br>P = 0.06 | -0.052(0.029); P = 0.08 | 2.028(0.814); P = 0.07 | **-0.89(0.43); P = 0.04** |
| Total Protein (g/dL) | -2.142(1.465); P = 0.148 | **-0.117(0.048 = 9); P<0.05** | 1.017(1.408); P = 0.472 | -0.51(0.64); P = 0.43 |
| AST (U/L) | -0.536(1.228); P = 0.663 | -0.064(0.041);<br>P = 0.12 | -1.057(1.168); P = 0.367 | **1.33(0.50); P = 0.01** |

[a]Wet Season(reference) is compared to Dry Season
[c]Male (reference) is compared to Female gender.

differentials. Similarly, in biochemical measurements, we also found the changes in reference intervals from the dry season to the wet season not statistically significant (P>0.05).

## Monthly and yearly pattern in parameters

We further analysed patterns based on changes in the monthly and yearly median of haematological and biochemical parameters (Figs 2, S1 and S2). These figures demonstrated a change in the pattern of our monthly median of haematological indices and biochemical measurements from January to December over five-year period. From these our study results (Figs 2, S1 and S2), we observed that laboratory parameters, including HB, RBC, HCT/PCV showed seasonal patterns throughout the year, with levels being normal to slightly low during the dry season and normal to high during the wet season. Likewise, we observed a seasonal pattern for

**Table 3.  The effect of exploratory seasonal factor on the reference ranges of the haematological indices and biochemical measurements (P>0.05).**

| Laboratory Parameters | Seasons | | | Dry Season | | Wet Season | |
|---|---|---|---|---|---|---|---|
| | n | Missing Data (n) | Median (Reference Range) | n | Median (Reference Range) | n | Median (Reference Range) |
| HB (g/dL) | 461 | 32 | 13.90(11.0–16.5) | 258 | 14.0(11.34–16.36) | 206 | 13.70 (10.70–16.29) |
| RBC[a](10^6/uL) | 444 | 49 | 4.87(3.90–5.85) | 238 | 4.95(4.06–5.89) | 201 | 4.77(3.77–5.76) |
| PLT (10^3/uL) | 433 | 60 | 244.5(158.6–346.2) | 238 | 237(156.9–338.2) | 197 | 248(163.5–361.1) |
| WBC (10^3/uL) | 437 | 56 | 5.40(3.52–7.65) | 245 | 5.34(3.51–7.60) | 194 | 5.34(3.52–7.80) |
| Neutrophils (10^3/uL) | 426 | 67 | 2.55(1.34–4.05) | 235 | 2.47(1.32–3.81) | 194 | 2.49(1.40–4.55) |
| Lymphocytes (10^3/uL) | 418 | 75 | 2.15(1.33–2.96) | 232 | 2.14(1.30–3.065) | 192 | 2.14(1.33–2.93) |
| Monocytes (10^3/uL) | 416 | 77 | 0.43(0.25–0.64) | 230 | 0.42(0.25–0.62) | 190 | 0.43(0.25–0.64) |
| Eosinophils (10^3/uL) | 335 | 158 | 0.13(0.13–0.31) | 201 | 0.14(0.02–0.36) | 138 | 0.11(0.01–0.26) |
| Basophils (10^3/uL) | 350 | 143 | 0.04(0.01–0.07) | 185 | 0.03(0.02–0.06) | 146 | 0.04(0.02–0.07) |
| Sodium (mmol/L) | 384 | 259 | 140(136–144) | 215 | 140(137–144) | | 140(136–144) |
| Potassium (mmol/L) | 498 | 145 | 4.12(3.6–4.7) | 235 | 4.20(3.6–4.7) | 152 | 4.10(3.70–4.50) |
| GGT (U/L) | 240 | 403 | 26(11.00–52.025) | 134 | 26(10.23–51.03) | 106 | 26(12.78–53.00) |
| Creatinine (umol/L) | 500 | 143 | 75.2(50.48–107.05) | 289 | 75(49.00–107.08) | 211 | 77(52.93–106.68) |
| Urea (mmol/L) | 507 | 136 | 3.1(1.7–4.9) | 381 | 3.1(1.80–4.7) | 217 | 3.1(1.70–4.9) |
| AST (U/L) | 354 | 289 | 20(12–32) | 211 | 20(12–34.5) | 138 | 20(12.43–29.00) |
| ALT (U/L) | 234 | 409 | 17(7.0–33.0) | 130 | 17(6.23–33.78) | 104 | 17(8.15–31.43) |
| Albumin (g/dL) | 236 | 407 | 40(34–47) | 150 | 40(34.00–47.00) | 82 | 40(34.03–45.00) |
| Total Protein (g/dL) | 336 | 307 | 76(67.00–83.63) | 195 | 76(68.00–85.00) | 147 | 76(67.00–83.35) |
| Globulin (g/dL) | 203 | 440 | 35(27.0–42.00) | 116 | 35(27–42.13) | 73 | 35(29.8–40.0) |

[a]Mean value was used for RBC parameter because the data was normally distributed.

total WBC, neutrophils, lymphocytes, and platelets from November to March with normal to slightly high median. In terms of biochemistry parameters, liver enzymes (AST, ALT, GGT, total protein) and potassium were generally high in the wet season (June to October) but moderately low in the dry season.

Finally, we assessed the changes in median values of the parameters from 2018 to 2022 using the boxplots in Figs 3 and 4. We followed this with a statistical significance test using the non-parametric Kruskal-Wallis test (H). Overall, no statistically significant difference was found (H = 0.1148, P>0.05) among the median values of all the haematological indices and biochemical measurements in the last five years (2018 to 2022). In haematological dataset, the median values for HB and RBC parameters decreased in 2020, while the median values for total WBC count, absolute neutrophils count, and platelets increased slightly. In the biochemical dataset, there was no clear pattern except for median values of AST and ALT, which showed a downward trend from 2018 to 2022.

## Discussion

The study examined the effects of seasonality on haematological indices, biochemical measurements and the reference ranges derived from our study datasets. Seasonal variations can be influenced by factors such as changes in environmental temperature and humidity, diet, physical activity, and seasonality of certain diseases like malaria. Overall, we found no statistically significant differences between the median values generated in the dry and wet seasons as well as reference intervals generated between the dry and wet season, indicating that there was no

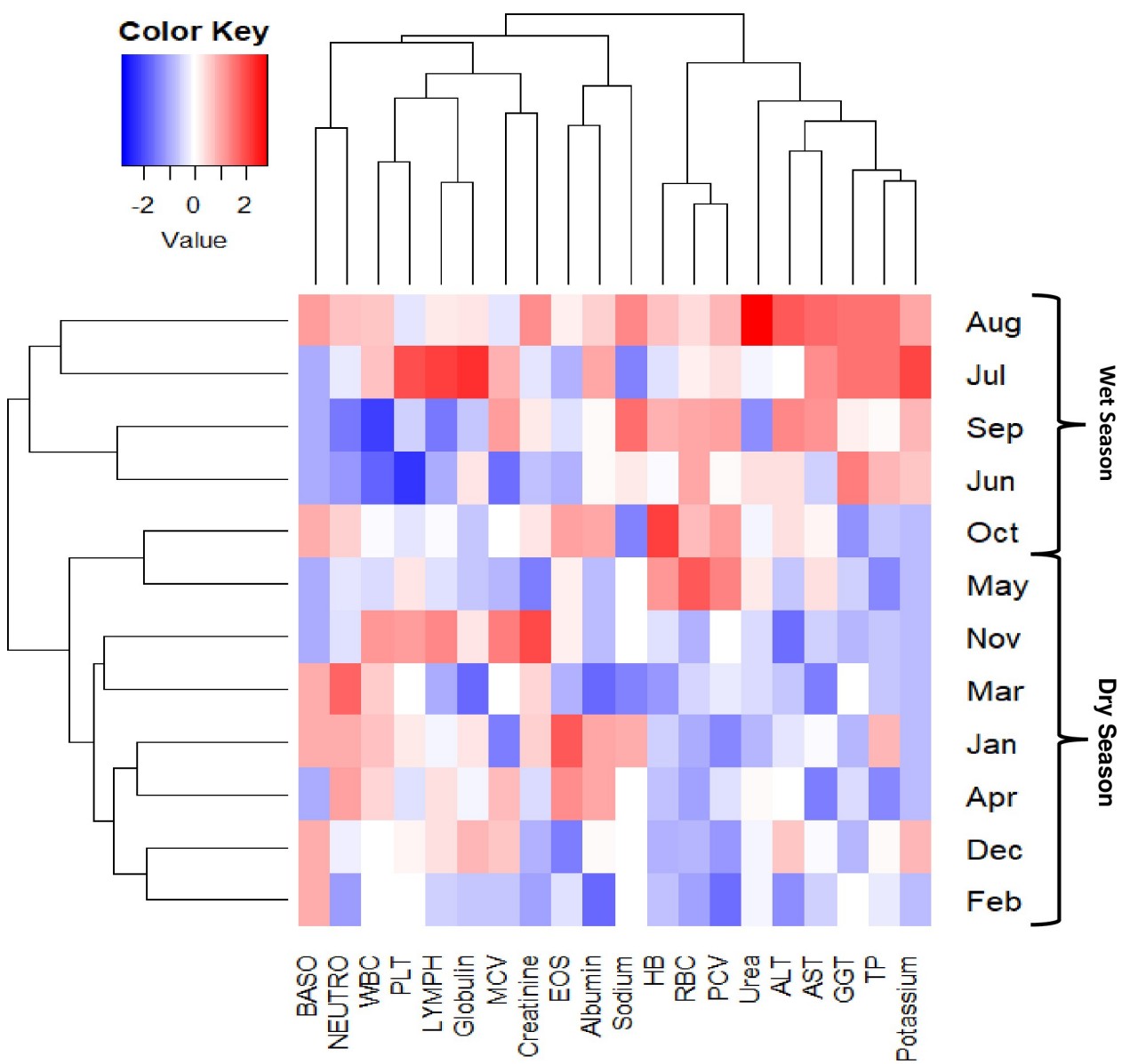

**Fig 2. Monthly pattern of haematological and biochemical parameters from 1st January 2018 to 31st December 2022 in five year period.** Hb; RBC; WBC; PLT; Neutrophils (NEUTRO); Lymphocytes (LYMPH); Mean cell volume (MCV); Packed cell volume (PCV); Eosinophils (EOS); Basophil (BASO); AST; ALT; GGT.

seasonal effect on the reference intervals derived from our study data. The reference intervals derived from our study data were comparable to the results reported by Adetifa et al in 2009, which focused on Gambian adults without any known underlying health conditions during a TB contact tracing study [17]. Notably, there was a slight difference in the upper limit of neutrophils reference intervals between the dry season (1.32–3.81) and wet season (1.40–4.55), suggesting the need for more informative reference intervals that consider seasonality. These seasonal variations should be taken into consideration, especially in research studies or clinical trials where patients are monitored at different times of the year. Being aware of these subtle seasonal variations can help prevent erroneous conclusions based on study data.

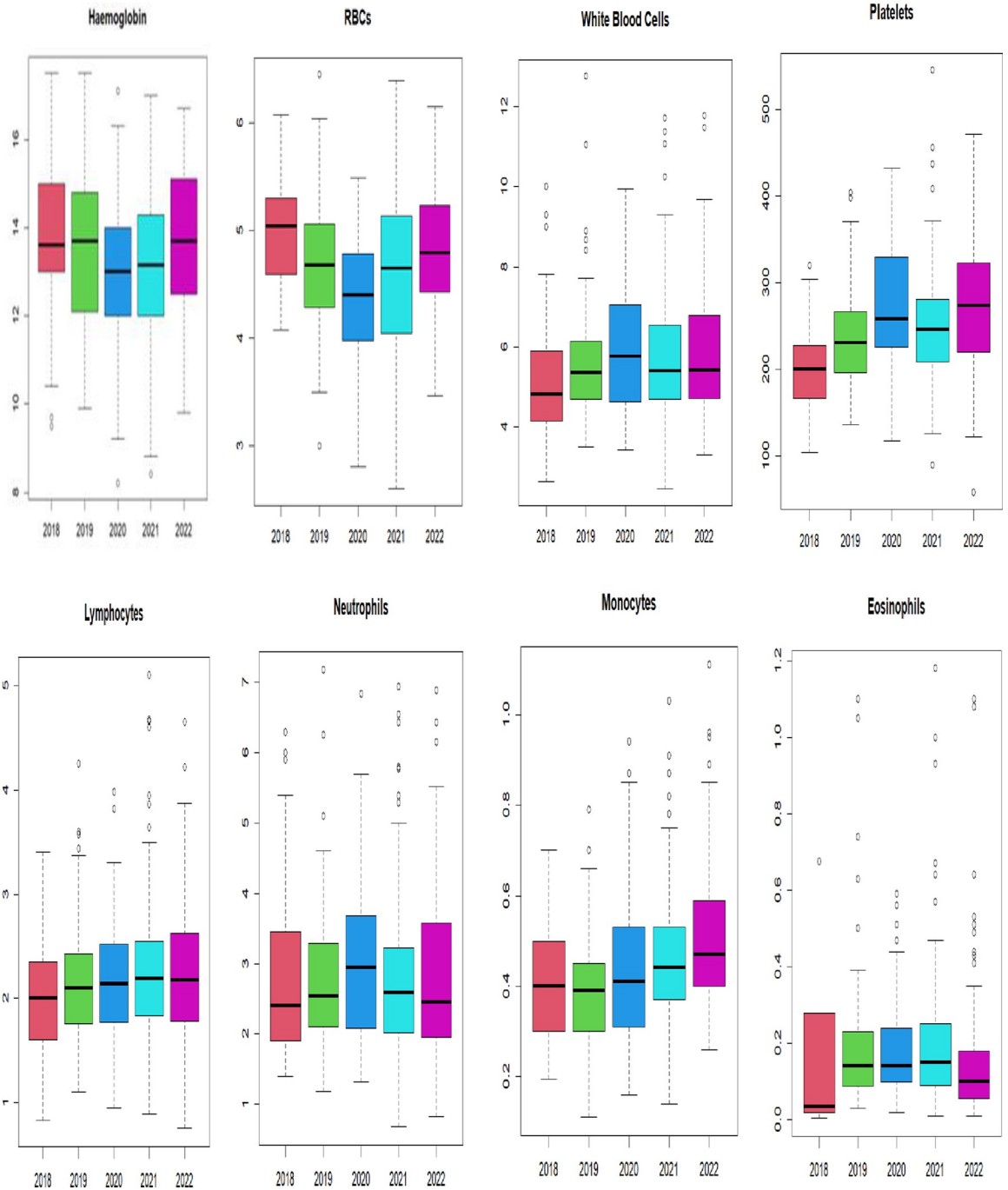

**Fig 3. Boxplot showing pattern/trend of haematological parameters in the last five years.** The results displayed in medians (Kruskal-Wallis chi-squared (H) = 0.063, P > 0.05).

To further investigate the seasonal variation pattern of these parameters from January to December, we observed moderately normal to slightly higher median pattern for haemoglobin and RBC during the wet season. Haemoglobin and RBC parameters are crucial for diagnosing anaemia in clinical settings. It has been reported that haemoglobin levels in individuals without any known underlying health conditions can vary due to seasonal factors such as changes

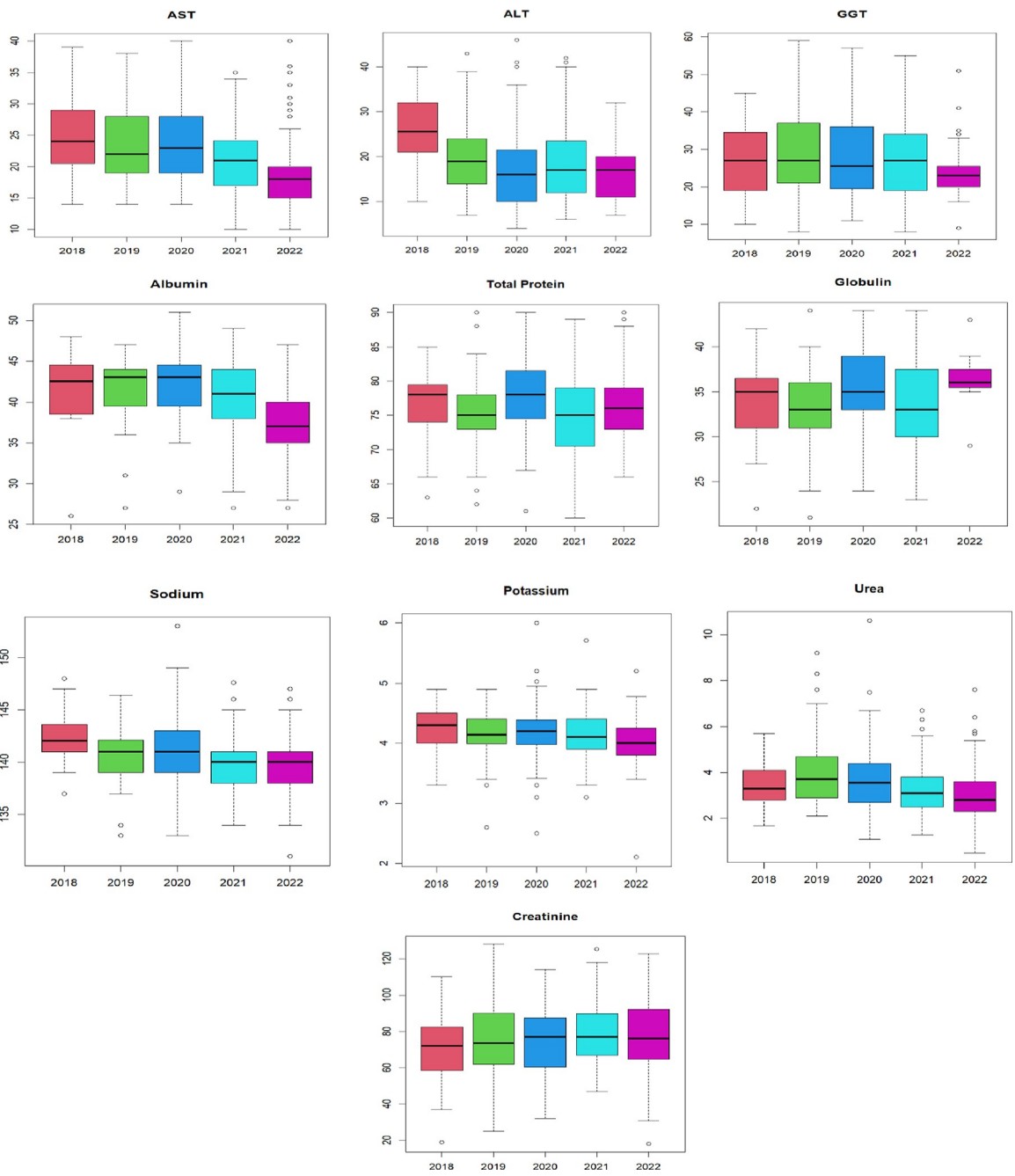

**Fig 4. Boxplot showing pattern/trend of biochemical parameters in the last five years.** (Kruskal-Wallis chi-squared (H) = 0.8433, P > 0.93).

in diet, including increased consumption of iron-rich foods during certain seasons, as well as asymptomatic infection [18]. In the Gambia, the wet season is characterized by the abundant availability of iron-rich foods, which could explain the higher levels of haemoglobin and RBCs during this period [13]. Besides, the moderately normal to high HCT seasonal variation patterns in the wet season commensurate with the high haemoglobin and RBC median values in the wet season, which could be attributed to low fluid intake during the wet season as a result of low temperature and humidity [13].

The seasonal variations in WBC and platelets were observed in our study from November to March. In the Gambia, changes in the environment during the wet season, such as increased rainfall and cold weather from November to March, favour the transmission of malaria parasites, which typically peaks in November, as well as respiratory infections [19–25]. The finding from our study supports previous findings regarding the seasonal effects on WBC and differentials [1]. Infectious diseases like malaria tend to flourish during the wet season, likely due to environmental factors such as heavy rainfall leading to stagnant water and waterlogged areas [26–29]. This can potentially lead to an increase in mean total WBC and differentials in the general population due to low-grade systemic inflammation in the community caused by malaria transmission and viral illnesses [3,10]. The seasonal variation in total WBC and differentials could be impacted by malaria parasite infection. However, our study records showed malaria microscopy was negative for all the haematological and biochemical datasets retrieved electronically.

Similar to haemoglobin and RBCs, the monthly pattern of liver enzymes and potassium changed based on seasonality. Our findings are aligned with earlier studies [30] that have demonstrated how seasonal changes in diet and fluid intake can impact biochemical measurements, especially electrolytes. The seasonal effects on these biochemical measurements may be attributed to the temperature difference between the wet and dry seasons [30–32]. A reference interval study conducted in Guinea also reported seasonal effects on biochemical parameters, particularly liver enzymes [27]. Despite reports of dehydration being more common during the dry season and potentially affecting biochemical measurements like serum creatinine and urea levels, our study found that median values for urea and creatinine remained relatively normal across both seasons [33]. Alarmingly, our study also revealed a consistent decline in the median values of AST and ALT from 2018 to 2022, warranting further investigation.

## Limitations

As observed in numerous retrospective studies, our study is also subject to limitations that restrict our control over certain factors. These factors include haemoglobin genotype status, body size, smoking, alcohol consumption, exercise, and contraceptive pill use, all of which have the potential to influence the outcome [34]. Additionally, our sample size was constrained by the available data, and we were unable to obtain additional data to enhance statistical power [35,36]. These limitations could be addressed through the implementation of a well-designed prospective study conducted on a general population, following the approach employed by Okebe et al. in 2016. Despite these limitations, we maintain that the results of our study are accurate, as our sample size was considerable. Therefore, our study remains representative and supports the current prospective reference intervals study that will provide a more comprehensive sampling and data stratification reflecting both dry and wet seasons.

## Conclusion

Our study found no significant seasonal effects on the reference intervals for haematological and biochemical parameters, despite the identification of certain parameters that exhibited some association with seasonality. However, we did observe seasonal variations in specific parameters such as haemoglobin, red blood cells, liver enzymes, and potassium levels from January to December. Furthermore, we did not observe significant yearly variations in the median values of all the haematological indices and biochemical measurements.

## Recommendations

Our findings emphasize the importance of considering seasonal variations in haematological indices and biochemical measurements when interpreting laboratory results and making

diagnostic decisions, as well as when recruiting participants for clinical trial projects. Our study should be followed by well-designed context-specific studies that employ standardized sampling and analytical methods within our geographical setting.

## Supporting information

**S1 Data. Study raw data.**
(XLSX)

**S1 Text. L2023.E03 Dibbasey Ethics Waiver 1.**
(PDF)

**S1 Fig. Haematological indices from January to December in five year period.** The point showed monthly medians in five year period.
(TIF)

**S2 Fig. Biochemical measurements from January to December in five year period.** The point showed monthly medians in five year period.
(TIF)

## Author Contributions

**Conceptualization:** Mustapha Dibbasey, Alfred Amambua-Ngwa.

**Data curation:** Mustapha Dibbasey, Mamudou Dahaba, Francess Sarfo, Rosyna Begum, Mustapha Kanteh, Nyima Sumareh, Mustapha Bakare, Solomon Umukoro, Alfred Amambua-Ngwa.

**Formal analysis:** Mustapha Dibbasey, Alfred Amambua-Ngwa.

**Funding acquisition:** Mustapha Dibbasey.

**Investigation:** Mustapha Dibbasey.

**Methodology:** Mustapha Dibbasey, Mamudou Dahaba, Francess Sarfo, Rosyna Begum, Mustapha Kanteh, Nyima Sumareh, Mustapha Bakare, Solomon Umukoro.

**Resources:** Mustapha Dibbasey.

**Supervision:** Alfred Amambua-Ngwa.

**Validation:** Mustapha Dibbasey, Alfred Amambua-Ngwa.

**Visualization:** Mustapha Dibbasey, Alfred Amambua-Ngwa.

**Writing – original draft:** Mustapha Dibbasey, Alfred Amambua-Ngwa.

**Writing – review & editing:** Mustapha Dibbasey, Mamudou Dahaba, Francess Sarfo, Rosyna Begum, Mustapha Kanteh, Nyima Sumareh, Mustapha Bakare, Solomon Umukoro, Alfred Amambua-Ngwa.

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
