## [Decision Letter · Decision Letter 0]

9 Apr 2024

PGPH-D-23-02498

Seasonal and yearly variations in haematological and biochemical parameters of healthy Gambian adults: a retrospective study

Dear Dr. Dibbasey,

Thank you for submitting your manuscript to PLOS Global Public Health. After careful consideration, we feel that it has merit but does not fully meet PLOS Global Public Health’s publication criteria as it currently stands. Therefore, we invite you to submit a revised version of the manuscript that addresses the points raised during the review process.

We look forward to receiving your revised manuscript.

Kind regards,

Valentina Gallo

Academic Editor

Journal Requirements:

1. Please ensure that Funding Information and Financial Disclosure Statement are matched.

2. In the Funding Information you indicated that no funding was received. Please revise the Funding Information field to reflect funding received.

3. Please provide separate figure files in .tif or .eps format only and remove any figures embedded in your manuscript file. Please also ensure all files are under our size limit of 10MB.

4. We have noticed that you have uploaded Supporting Information files, but you have not included a list of legends. Please add a full list of legends for your Supporting Information files after the references list.

5. We notice that your supplementary files are included in the manuscript file. Please remove them and upload them with the file type 'Supporting Information'. Please ensure that each Supporting Information file has a legend listed in the manuscript after the references list.

6. In the online submission form, you indicated that "The data for the study is fully available upon request". All PLOS journals now require all data underlying the findings described in their manuscript to be freely available to other researchers, either 1. In a public repository, 2. Within the manuscript itself, or 3. Uploaded as supplementary information.

Additional Editor Comments (if provided):

Reviewers' comments:

Reviewer's Responses to Questions

**Comments to the Author**

1. Does this manuscript meet PLOS Global Public Health’s publication criteria? Is the manuscript technically sound, and do the data support the conclusions? The manuscript must describe methodologically and ethically rigorous research with conclusions that are appropriately drawn based on the data presented.

Reviewer #1: Yes

Reviewer #2: No

2. Has the statistical analysis been performed appropriately and rigorously?

Reviewer #1: Yes

Reviewer #2: No

3. Have the authors made all data underlying the findings in their manuscript fully available (please refer to the Data Availability Statement at the start of the manuscript PDF file)?

Reviewer #1: Yes

Reviewer #2: Yes

4. Is the manuscript presented in an intelligible fashion and written in standard English?

Reviewer #1: No

Reviewer #2: No

5. Review Comments to the Author

Reviewer #1: The study requires major modification. The use of the term/word healthy people is problematic. If someone came to the laboratory for routine check-up/testing does not make such individual healthy person. Perhaps, individual with no known underlying health condition could suffice. Although the study design was retrospective, reading through the entire write-up appears more prospective.

Reviewer #2: R1:

Title:

-if possible; the title can be changed by remove of the word (yearly), and addition of the year to (retrospective study 2018-2022)

Abstract:

-Divide background rom the aim of the study

-Better to write the objective of the study.

-in abstract give brief idea about study design, study population and area, sampling, methods of data collection and the measurments (not more than 5 rows).

-briefly give the main results of the study with the accurate numbers.

-there is some confusion rgarding the effect of seasonal differences in the measurement of hematological index and biochemical measurements.

-write only the conclusion of the result. interpretation of result can be discussed in the section of discussion.

-use mean full and brief key words

Methods:

- some texts were not needed

- regarding exclusion and inclusion criterias, what about the other seasonal infections e.g. influenza

- what is the final diagnose of the healthy participants who participate in the current study.

- the biomedical history of the participants is absent from the current study.

- inclusion and exclusion criteria should minimize the un-suspected results or errors.

Results:

-493 participant for haematological indexes.

-643 participant for biochemical measurements.

-the measured parameters was for separated individuals?

-643 and 493 individuals measured for different parameters.

-the current study have 2 study population.

-the study must divide into two studies.

In the results section its clearly that there was 2 study population:

-in the first one the researcher examine the seasonal haematological changes.

-in the second one the researcher examine the biochemical changes.

Then:

-there was a comparisons between the two populations.

So:

-the researcher must find the haematological indexes and biochemical measurement change in one population.

OR:

-compare the change in hematological indexes in the first population which occurs as a result of seasonal change as separate work.

-then find the seasonal changes in biochemical measurements in the second population

-finally the researcher can compare the results of the two studies

Discussion:

-in discussion there is unacceptable justification regarding the personal, socio- demographic and life style data which didn't appear in the methods of data collection in form of questionnaire or study objectives.

6. PLOS authors have the option to publish the peer review history of their article (what does this mean?). If published, this will include your full peer review and any attached files.

**Do you want your identity to be public for this peer review?** For information about this choice, including consent withdrawal, please see our Privacy Policy.

Reviewer #1: **Yes: **David Larbi Simpong

Reviewer #2: No

---

## [Decision Letter · Decision Letter 1]

22 May 2024

PGPH-D-23-02498R1

Seasonal variations in haematological and biochemical parameters of healthy Gambian adults: Retrospective Study 2018-2022

Dear Dr. Dibbasey,

Thank you for submitting your manuscript to PLOS Global Public Health. After careful consideration, we feel that it has merit but does not fully meet PLOS Global Public Health’s publication criteria as it currently stands. Therefore, we invite you to submit a revised version of the manuscript that addresses the points raised during the review process.

Please consider carefully the notes from one of the reviewers and double check your figures, and transparently disclose the missing values leading to different totals in your tables. Consider adding a flowchart if needed.

We look forward to receiving your revised manuscript.

Kind regards,

Valentina Gallo

Academic Editor

Journal Requirements:

Additional Editor Comments (if provided):

Reviewers' comments:

Reviewer's Responses to Questions

**Comments to the Author**

1. If the authors have adequately addressed your comments raised in a previous round of review and you feel that this manuscript is now acceptable for publication, you may indicate that here to bypass the “Comments to the Author” section, enter your conflict of interest statement in the “Confidential to Editor” section, and submit your "Accept" recommendation.

Reviewer #1: All comments have been addressed

Reviewer #2: All comments have been addressed

2. Does this manuscript meet PLOS Global Public Health’s publication criteria? Is the manuscript technically sound, and do the data support the conclusions? The manuscript must describe methodologically and ethically rigorous research with conclusions that are appropriately drawn based on the data presented.

Reviewer #1: Yes

Reviewer #2: No

3. Has the statistical analysis been performed appropriately and rigorously?

Reviewer #1: Yes

Reviewer #2: No

4. Have the authors made all data underlying the findings in their manuscript fully available (please refer to the Data Availability Statement at the start of the manuscript PDF file)?

Reviewer #1: Yes

Reviewer #2: Yes

5. Is the manuscript presented in an intelligible fashion and written in standard English?

Reviewer #1: Yes

Reviewer #2: No

6. Review Comments to the Author

Reviewer #1: All the comments have been addressed

Reviewer #2: Unacceptable data analysis.

7. PLOS authors have the option to publish the peer review history of their article (what does this mean?). If published, this will include your full peer review and any attached files.

**Do you want your identity to be public for this peer review?** For information about this choice, including consent withdrawal, please see our Privacy Policy.

Reviewer #1: **Yes: **David Larbi Simpong

Reviewer #2: **Yes: **Nahla Ahmed Mohammed Abderhman

---

## [Decision Letter · Decision Letter 2]

5 Aug 2024

PGPH-D-23-02498R2

Seasonal variations in haematological and biochemical parameters of healthy Gambian adults: Retrospective Study 2018-2022

Dear Dr. Dibbasey,

Thank you for submitting your manuscript to PLOS Global Public Health. After careful consideration, we feel that it has merit but does not fully meet PLOS Global Public Health’s publication criteria as it currently stands. Therefore, we invite you to submit a revised version of the manuscript that addresses the points raised during the review process.

Reviewer 2 has provided some additional comments in the attached file - the comments are also found below, but in the attachment you can see the location where they are making the comments and suggestions.

We look forward to receiving your revised manuscript.

Kind regards,

Hanna Landenmark

Staff Editor

Journal Requirements:

Additional Editor Comments (if provided):

Reviewers' comments:

Reviewer's Responses to Questions

**Comments to the Author**

1. If the authors have adequately addressed your comments raised in a previous round of review and you feel that this manuscript is now acceptable for publication, you may indicate that here to bypass the “Comments to the Author” section, enter your conflict of interest statement in the “Confidential to Editor” section, and submit your "Accept" recommendation.

Reviewer #1: All comments have been addressed

Reviewer #2: (No Response)

2. Does this manuscript meet PLOS Global Public Health’s publication criteria? Is the manuscript technically sound, and do the data support the conclusions? The manuscript must describe methodologically and ethically rigorous research with conclusions that are appropriately drawn based on the data presented.

Reviewer #1: Yes

Reviewer #2: No

3. Has the statistical analysis been performed appropriately and rigorously?

Reviewer #1: Yes

Reviewer #2: No

4. Have the authors made all data underlying the findings in their manuscript fully available (please refer to the Data Availability Statement at the start of the manuscript PDF file)?

Reviewer #1: Yes

Reviewer #2: Yes

5. Is the manuscript presented in an intelligible fashion and written in standard English?

Reviewer #1: Yes

Reviewer #2: Yes

6. Review Comments to the Author

Reviewer #1: The concerns raised have been addressed by authors

Reviewer #2: R2:

1. Regarding study participants, grouping and measurements; are still not clear

2. no need to mention type of analysis in the results in abstract

3. not important (ignore this and focus on the sample that meet research criteria)

4. focus on the objectives

5. in the first group the researcher measure hematological parameters

6. in the first group the researcher measure hematological parameters

7. mention number of participant who were in the median

8. mention number of participant who were in the mean

9. table 2 is for un-desirable date please no need for table 2 as mentioned above

10. a lot of repeated data and information

11. add column for the missing data in all hematological and biochemical measurements

12. how you compare the hematological measurements with biochemical measurements

13. use only the abbreviations for all the define term

discussion:

14. you mean (using reference interval as standared?)

15. add the subtitle (limitations and recommondations)

7. PLOS authors have the option to publish the peer review history of their article (what does this mean?). If published, this will include your full peer review and any attached files.

**Do you want your identity to be public for this peer review?** For information about this choice, including consent withdrawal, please see our Privacy Policy.

Reviewer #1: No

Reviewer #2: **Yes: **Nahla Ahmed Mohammed Abderhman

---

## [Decision Letter · Decision Letter 3]

26 Aug 2024

Seasonal variations in haematological and biochemical parameters of healthy Gambian adults: Retrospective Study 2018-2022

PGPH-D-23-02498R3

Dear Dibbasey,

We are pleased to inform you that your manuscript 'Seasonal variations in haematological and biochemical parameters of healthy Gambian adults: Retrospective Study 2018-2022' has been provisionally accepted for publication in PLOS Global Public Health.

Best regards,

Julia Robinson

Executive Editor

Reviewer Comments (if any, and for reference):

Reviewer's Responses to Questions

**Comments to the Author**

1. If the authors have adequately addressed your comments raised in a previous round of review and you feel that this manuscript is now acceptable for publication, you may indicate that here to bypass the “Comments to the Author” section, enter your conflict of interest statement in the “Confidential to Editor” section, and submit your "Accept" recommendation.

Reviewer #1: All comments have been addressed

Reviewer #2: All comments have been addressed

2. Does this manuscript meet PLOS Global Public Health’s publication criteria? Is the manuscript technically sound, and do the data support the conclusions? The manuscript must describe methodologically and ethically rigorous research with conclusions that are appropriately drawn based on the data presented.

Reviewer #1: Yes

Reviewer #2: Yes

3. Has the statistical analysis been performed appropriately and rigorously?

Reviewer #1: Yes

Reviewer #2: Yes

4. Have the authors made all data underlying the findings in their manuscript fully available (please refer to the Data Availability Statement at the start of the manuscript PDF file)?

Reviewer #1: Yes

Reviewer #2: Yes

5. Is the manuscript presented in an intelligible fashion and written in standard English?

Reviewer #1: Yes

Reviewer #2: No

6. Review Comments to the Author

Reviewer #1: All concerns raised has been addressed by authors

Reviewer #2: R3:

Thank you for your patience.

General comments:

Great effort is needed to accomplish this work.

Please see the enclosed document for more information

*avoid to use pronouns like (we, you, our…etc)

The action you must be use is in past participle

Avoid repeated information

Please down load scientific peer reviewed published paper related to your current research to mimic their scientific language.

Abstract:

abstract is presented in clear way. fine.

key words are too much. (suggestion)=seasonal variation, hematological indices, biochemical measurements, Gambia

Methodology:

study setting: please mention the name of the lab first then its quality and type of routine work.

author can give brief description of lab quality without evidence.

mention the source of ethical approval and consent only.

Study design: missing date will appear in analysis. no need for justification

Results:

Some references and researchers express p-value<=0.05

Suggestion to results section:

1. * write all results regarding hematological parameters, then in the next paragraph write about biochemical parameters

2. *then compare all the results(if needed)

3. *comparison become uninteresting.

please mention the same source (eg. table 1) in one paragraph one time. delete repeated information

Another suggestion:

Seasonal and Yearly Effects and Reference Intervals:

1. compare hematological parameters only in wet and dry seasons using different methods of data analysis.

2. compare biochemical parameters only in wet and dry seasons using different methods of data analysis.

Monthly and Yearly Pattern in Parameters:

suggestions to this section:

1. *the author write about all changes that occurs over all year per month, So, this results is a base of the over-mentioned results.

2. *write this paragraph firstly then put the comparisons of wet and dry seasons of hematological and biochemical parameters by different methods of analysis.

3. *that mean you need to rearrange results section.

Discussion:

repeated information: author can justify and present the results in different style.

Repetition of same information by similar words is so dull.

Justification of variation on hematological indices as result of affecting by malaria parasite which showed negative results means :

1. previous effect with malaria parasite result in TWBCs, or

2. test for malaria parasite is false, or

3. affect with malaria parasite did not result in the seasonal variation of TWBCs

limitations: re- write in points. put limitations in points without comparing with other study.

Recommendations: re- write in points

Conclusion: need revision

7. PLOS authors have the option to publish the peer review history of their article (what does this mean?). If published, this will include your full peer review and any attached files.

**Do you want your identity to be public for this peer review?** For information about this choice, including consent withdrawal, please see our Privacy Policy.

Reviewer #1: No

Reviewer #2: **Yes: **Nahla Ahmed Mohammed Abdelrahman
